# Enhancing Intangible Cultural Heritage for Sustainable Tourism Development in Rural Areas: The Case of the "Marche Food and Wine Memories" Project (Italy)

**Mara Cerquetti** [1,*], **Concetta Ferrara** [1] , **Annamaria Romagnoli** [1] **and Gianluca Vagnarelli** [2]

1    Department of Education, Cultural Heritage and Tourism, University of Macerata, 62100 Macerata, Italy
2    i-strategies, 63073 Offida, Italy
*    Correspondence: mara.cerquetti@unimc.it

**Abstract:** In the context of increasing interest in the contribution made by culture to the implementation of the goals and targets of the 2030 Agenda for Sustainable Development, the present research investigates how intangible cultural heritage (ICH) can help sustainable tourism in rural areas. Adopting a case study methodology, we analyzed the "Marche Food and Wine Memories" project, an initiative promoted by CiùCiù, a winery based in Offida, a small village in the Marche region (Italy). After discussing the strategies and tools adopted to enhance rural heritage, the analysis focuses on the involvement of local communities and businesses in the different phases of the process. The research aimed to understand: (1) the project's current contribution to the economic, social, cultural and environmental dimensions of sustainability; and (2) its strengths and weaknesses and possible future improvements. The research findings confirm the high potentialities of ICH-based initiatives for sustainable tourism development in rural areas, but also reveal the need to improve the level of networking with local businesses and highlight gaps in marketing and management skills. Finally, the results provide policy and managerial implications for similar ICH-based initiatives.

**Keywords:** rural heritage; sustainable development; tourism; food and wine; sharecropping



## 1. Introduction

The European Union's long-term vision for rural areas up to 2040 highlights their potential and importance for European identity. They are rich in environmental and cultural resources and can attract a new and increasing, but still unmet, demand for culture from those seeking a broad and deep experience of place-specific resources, such as crafts and gastronomy, traditional knowledge and practices [1]. However, in the current context, rural areas are also experiencing population decline, an ageing population, economic decline, difficulty accessing health and education services and limited digital connectivity [2]. As already highlighted by UNESCO [3], intangible cultural heritage (ICH) could be a powerful lever for promoting sustainable development in these areas. On the one hand, it can attract tourists at national, regional and international levels and boost local businesses (economic sustainability); on the other, it contributes to well-being and quality of life and can strengthen social cohesion and inclusion (social sustainability). Moreover, it helps protect biodiversity and enhance environment-friendly local knowledge and practices, e.g., the sustainable use of natural resources (environmental sustainability). Finally, it contributes to knowledge sharing and increases human capital (cultural sustainability) [4]. However, this topic has not been specifically analyzed in the scientific literature on sustainable tourism development in rural areas.

In this context, the present research discusses the contribution of rural heritage to the four dimensions of sustainability for tourism development in rural areas. Adopting a case study methodology [5,6], we investigated the "Marche Food and Wine Memories" project, an initiative promoted by CiùCiù, a winery based in Offida, a small village in the Marche

region (Italy). The project aims to safeguard and enhance the rural cultural heritage related to food and wine in the southern Marche region by collecting the oral memories of the last witnesses of *mezzadria* (sharecropping). The project also adopts storytelling methods and includes guided and gamified tours around sharecropping houses where the collected stories are presented to visitors.

After discussing strategies and tools adopted to enhance rural heritage, our analysis focused on the involvement of local communities and businesses in the different phases of the project. The literature review on the role of tourism and cultural heritage in sustainable development in rural areas allowed us to build a conceptual framework that guided and structured the analysis of the case study. The research aimed to understand:

- The project's current contribution to the economic, social, cultural and environmental dimensions of sustainability;
- Its strengths and weaknesses and possible future improvements.

The research findings confirm the high potentialities of ICH-based initiatives for the sustainable tourism development of rural areas, but also reveal the need to improve the level of networking with local businesses. Moreover, they highlight gaps in the marketing and management skills needed to ensure long-term sustainable development. Finally, the results provide policy and managerial implications for similar ICH-based initiatives.

The paper is structured as follows: Section 2 presents the theoretical background through a critical discussion of the scientific literature on the role of tourism and cultural heritage in promoting sustainable development in rural areas. Section 3 presents the research methodology and Section 4 the research results; and Section 5 discusses the main findings and their impact on museum management. Conclusions are drawn in Section 6, which point out the policy and managerial implications, research limitations and suggest further research.

## 2. Theoretical Background

### 2.1. Sustainability and Tourism Development

Meeting "the needs of the present without compromising the ability of future generations to meet their own needs" [7] is still one of the most urgent challenges contemporary society faces when looking for common solutions to the planet's biggest problems.

In view of its considerable role in generating prosperity, improving livelihoods and preserving the environment, tourism is in a special position to drive sustainable development [8]. First, thanks to its dynamic nature and impact on the economies of many countries and destinations, it can be a driver of economic development at local, regional and national levels [9]. Moreover, thanks to the interaction between tourists, local communities and local environments, it can be a source of knowledge and awareness and make visitors and hosts more conscious of environmental issues and the differences between nations and cultures [10].

At the global level, this crucial role is also confirmed by the recurring reference to tourism in key United Nations (UN) documents on sustainable development, such as *The Future We Want* [11], the *Addis Ababa Action Agenda on Financing for Development* [12] and the *2030 Agenda for Sustainable Development* [13]. The last of these committed all countries to pursue a set of 17 Sustainable Development Goals (SDGs) that would lead to a better future.

In the UN 2030 Agenda framework, tourism has been accorded a transversal function since it indirectly contributes to achieving all SDGs and is included as a specific target in three of them. First, tourism is recognized as one of the driving forces of global economic growth (SDG 8) since it currently accounts for 1 in 11 jobs worldwide. In addition, when giving access to decent work opportunities in the tourism sector, society can benefit from increased skills and professional development. Consequently, by 2030, specific policies are to be implemented that will promote sustainable tourism, create jobs and promote local culture and products. Second, tourism can act as a lever to promote sustainable consumption and production (SDG 12). Third, tourism should stimulate the sustainable

use of marine resources and serve as a vehicle to promote the blue economy by supporting the conservation and preservation of fragile marine ecosystems (SDG 14).

In this context, the UN General Assembly's designation of 2017 as the International Year of Sustainable Tourism for Development established another milestone by setting the scene for a stronger awareness among public and private sector decision-makers and the public about sustainable tourism's contribution to development [14]. Along this journey, *Tourism and the Sustainable Development Goals. Journey to 2030*, the book published by the UN World Tourism Organization (UNWTO) and the UN Development Programme (UNDP), clarified the links between tourism and the SDGs, and provided policy recommendations to accelerate the shift towards a more sustainable tourism sector by aligning policies, business operations and investments with the SDGs and making tourism a catalyst for positive change [15].

The relationship between tourism and sustainable development can be both positive and negative. On the one hand, tourism can be a source of opportunities in terms of entrepreneurship [16], employability [17] and attracting investments. It can also create tangible economic value for natural and cultural resources and therefore bring general economic growth [18,19]. From a social perspective, tourism benefits are also related to community pride, tolerance and a stronger sense of ethnic identity [20–22]. On the other hand, tourism can contribute to the erosion of scarce resources and local and global pollution, and can compromise fragile ecosystems causing the degradation of the physical environment and bringing disruption to wildlife [23]. Tourism can also exert negative pressures on host communities [24–26]. Finally, it can be an unstable source of income, as it is often susceptible to actual or perceived changes in the environmental and social conditions of destinations [10].

In this perspective, sustainable tourism, as a form of tourism addressing the needs of visitors, the tourism industry, the environment and host communities, is expected to create a balance among environmental, economic and socio-cultural impacts. Several actions should be implemented, such as promoting the optimal use of environmental resources, respecting the authenticity of places and local communities, supporting the conservation of cultural heritage and traditional values, as well as ensuring long-term economic operations, including stable employment, income-earning opportunities and social services [27]. In this perspective, the United Nations Environment Programme (UNEP) and the World Tourism Organization (UNWTO) released an Agenda for Sustainable Tourism [10], consisting of a set of 12 specific and interrelated aims addressing the economic, social and environmental impacts of tourism (Table 1). The agenda provides a framework for developing policies for more sustainable tourism, minimizing the negative effects of tourism on society and the environment and maximizing tourism's positive and creative contribution to local economies, the conservation of natural and cultural heritage and quality of life for hosts and visitors.

**Table 1.** The 12 aims of the Agenda for Sustainable Tourism. Source: own elaboration on [10] (pp. 18–19).

| Aim | Definition |
| --- | --- |
| **Economic viability** | Ensuring the viability and competitiveness of tourism destinations and enterprises. |
| **Local prosperity** | Maximizing the contribution of tourism to the economic prosperity of the host destination. |
| **Employment quality** | Strengthening the number and quality of local jobs created and supported by tourism. |
| **Social equity** | Seeking a widespread and fair distribution of economic and social benefits from tourism throughout the recipient community. |
| **Visitor fulfilment** | Providing a safe, satisfying and fulfilling experience for visitors without any discrimination. |
| **Local control** | Engaging and empowering local communities in planning and decision-making about the management and future development of tourism in their area. |

Table 1. *Cont.*

| Aim | Definition |
| --- | --- |
| **Community well-being** | Maintaining and strengthening the quality of life in local communities, avoiding any form of social degradation or exploitation. |
| **Cultural richness** | Respecting and enhancing the historic heritage, authentic culture, traditions and distinctiveness of host communities. |
| **Physical integrity** | Maintaining and enhancing the quality of landscapes, avoiding the physical and visual degradation of the environment. |
| **Biological diversity** | Supporting the conservation of natural areas, habitats and wildlife, and minimizing damage to them. |
| **Resource efficiency** | Minimizing the use of scarce and non-renewable resources in the development and operation of tourism facilities and services. |
| **Environmental purity** | Minimizing the pollution of air, water and land and the generation of waste by tourism enterprises and visitors. |

*2.2. Sustainable Tourism Development in Rural Areas*

Achieving sustainable development depends on an area's territorial, economic and social idiosyncrasies. The implications can differ between populated and developed urban areas, and marginal and disadvantaged areas. To this aim, in recent years, European regional policies and structural funds have focused on the need to identify new innovation-driven models to promote regional development and economic growth and reduce territorial imbalances in different areas [28,29].

These challenges are considerable in rural areas. These portions of territory are affected by weakening agricultural or soft industrial activities, a lack of services, low living standards and out-migration [30]. However, they can also rely on place-specific tangible and intangible cultural resources and meet post-modern tourist needs, such as the quest for authenticity and revival of cultural roots through experience-based interactions with local culture and immersion in the everyday local life [31,32].

The concept of authenticity in tourism was introduced many decades ago by MacCannell [33] and has increasingly become a crucial topic for scholars, who have focused on many aspects [34–36], including the risks related to ambiguity and limited authenticity in tourism [37,38]. More recently, some researchers have highlighted the existence of a positive connection between the perception of a place's authenticity, attachment to a place and defining a feeling of loyalty for a destination [39,40]. In this perspective, rural tourism—defined as "a type of tourism activity in which the visitor's experience is related to a wide range of products generally linked to nature-based activities, agriculture, rural lifestyle/culture, angling and sightseeing" [41]—may be a source of development for rural areas and provide authentic experiences for tourists.

Indeed, this form of tourism can be an additional economic activity whereby rural areas no longer have to depend entirely on primary activities such as agriculture and livestock [42]. Moreover, this form of tourism could play a crucial role in sustainable development [43] and act as a significant factor in socio-economic development and regeneration [44]. Specifically, by supporting the protection of local natural and cultural capital and by using it sustainably, rural tourism can establish a balance between the economic and ecological dimensions of development and achieve economic, environmental and socio-cultural growth [45–47].

The scientific literature review on the topic identifies several positive and negative externalities of tourism in rural areas. First, many scholars focus on the nature of rural tourism as a vehicle for the economic development of rural areas, in terms of economic growth and diversification [48,49], but also as a driver for agricultural development [50] and a stimulus for the creation or growth of new local enterprises [51]. Some authors also highlight the influence of rural tourism in reducing outgoing migration [52] and dealing

with the challenge of depopulation [53] through a mechanism of population retention [36]. Indeed, studies show how tourism activities can help increase revenues, employability and job creation, and modernize facilities [53–55]. When considering the social dimension of tourism for sustainable development in rural areas, some studies stress the capacity of this type of tourism to improve socio-economic well-being [56,57] and the overall quality of life of residents [58,59]. Other studies demonstrate that rural tourism can bring tangible economic value to natural and cultural resources, encouraging residents to preserve local heritage through specific actions, thus increasing a local sense of pride and belonging [54] and the quality of the tourist experience [60–62].

Focusing on the negative impact of tourism activities on rural areas, scholars point out the inability of rural tourism to make a significant contribution in terms of new jobs and increasing the quality of life of local people [63]. In some cases, there are also negative environmental and socio-cultural consequences, such as the destruction of natural resources (e.g., local vegetation and landscape), the loss of traditional culture, the rise of social conflicts between hosts and guests [64] and a failure to involve local communities in the planning process [65]. When investigating the economic impact, the literature also shows the risks of an unbalanced distribution of economic benefits and higher prices for certain goods and services [66].

In this context, measuring the impact of tourism is essential for preventing conflicts between tourism and sustainability and for making tourism an effective tool for the development of rural areas [67].

### 2.3. The Role of Cultural Heritage for Sustainable Development

Over the last fifteen years, the international framework provided by the Council of Europe Convention on the Value of Cultural Heritage for Society, signed in Faro in 2005 (Italy signed the Faro Convention on 27 February 2013 and ratified it on 15 December 2020. The Convention entered into force on 1 April 2021), has significantly broadened the concept of cultural heritage, opening new opportunities for its use and contribution to sustainable development.

The Faro Convention finally recognized the open, relational and dynamic nature of cultural heritage as the product of the continuous and changing interaction between people and places: not only *heritage-as-object*, but also *heritage-as-process* [68]. Furthermore, according to the Convention, cultural heritage can be considered a two-faced Janus; that is, both a *source* and a *resource*. On the one hand, it is evidence of the past and documents the origins of Europe, "a shared source of remembrance, understanding, identity, cohesion and creativity" (Article 3(a)). On the other, it is a resource from which we can draw cultural, social and economic benefits for the future (Preamble; Article 2(a); Article 7(c)) and sustainable development (Articles 1, 3 and 5). From previous research on cultural heritage [69,70], we can summarize the paradigm shift introduced by the Faro Convention as a complete and profound reversal: of authority (from top to bottom), of the object (from the exceptional to everything), of value (from the value itself to use-value) and, therefore, of purpose (from "museification" to enhancement) [71].

When analyzing the value of cultural heritage, Montella [72] identified three different types of use-value that can be drawn from cultural heritage: *presentation value*, as an information-based value, corresponding to the communication of the meanings of cultural heritage, i.e., authentic, aesthetic, symbolic, social, historical and documentary value [73]; *landscape value*, as a systemic value concerning the safeguard of the environment and territory through environmental policies and city planning [74]; *production value*, referring to market uses flowing from cultural heritage and profit for businesses operating in different sectors, e.g., restoration, publishing, tourism, construction, real estate business, performing arts, etc. [75]. If cultural heritage is stratified through time in a specific context, it can influence a firm's value creation and competitive advantage.

As shown in Figure 1, these three values are intrinsically connected to sustainability. Presentation value targets cultural sustainability, while landscape value can generate

both environmental sustainability, in terms of environment and landscape quality, and social sustainability, by improving quality of life. Finally, production value contributes to economic sustainability. Thus, the proper use of cultural heritage can trigger a virtuous sustainable cycle. Increasing numbers of citizens with a better understanding of cultural heritage and its environment (*cultural sustainability*) "leads to people valuing it more and as a consequence caring for it better" [76], thus improving the environment and landscape quality (*environmental sustainability*) and promoting quality of life and well-being (*social sustainability*) [77]. Moreover, the enhancement of cultural heritage can create job opportunities and attract tourists, making it a source of income for the cultural and creative industries, the tourism and hospitality sectors, etc. (*economic sustainability*).

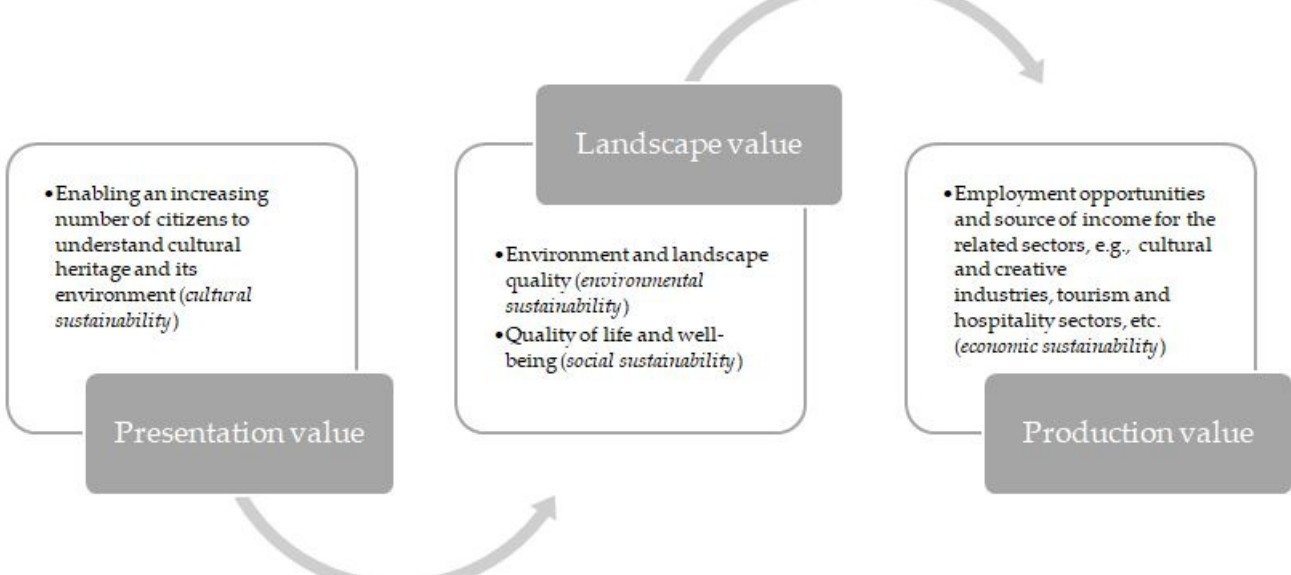

**Figure 1.** Cultural heritage value and sustainable development. Source: own elaboration.

This virtuous cycle is particularly effective in rural areas, where heritage-based tourism can play a crucial role in counteracting population and economic decline and the abandonment and disappearance of cultural heritage. Indeed, considering its connections with the productive vocation of the territory and its potential as an authentic local asset, cultural heritage can contribute to the attractiveness and distinctiveness of a rural destination [78]. Specifically, intangible cultural heritage (ICH), including all practices, traditions and knowledge that a community considers part of its cultural heritage [79], can crucially contribute to increasing awareness of local cultural specificity [80] and allow a rural destination to better meet the experience needs of tourists.

These assumptions become more relevant in the scenarios defined by the COVID-19 pandemic, which had a devastating and disruptive effect on travel and tourism, as well as on the hospitality-, arts- and events-affiliated sectors [81]. Indeed, destination crisis events have negative impacts on tourists' emotions, attitudes and behaviors and thus influence sustainable tourism development [82,83]. Nevertheless, although the COVID-19 epidemic over the last two years has prevented tourists from traveling, tourists still expect to experience pure, authentic and unrestricted tourist destinations [40]. In this perspective, rural areas, providing authentic, uncrowded and in-touch-with-nature touristic experiences, have been discovered as one of the most promising places for the development of domestic tourism [84], but also as a key element for "revenge tourism", providing a chance for a degrowth-oriented restart that forms the foundation for a more sustainable tourism sector [85].

Many possible benefits can be identified, while taking into account the analysis carried out in Section 2.1 (Table 2).

**Table 2.** The four dimensions of sustainability in heritage-based tourism in rural areas. Source: own elaboration.

**ECONOMIC SUSTAINABILITY**
[86–90]

1. Source of income for the tourism and hospitality sectors.
2. Source of income for the tourism and hospitality-related sectors.
3. Encouragement for local entrepreneurship.
4. Source of attraction for external investment.
5. Employment opportunities for residents.

**SOCIAL SUSTAINABILITY**
[88–93]

1. Enhancement of quality of life.
2. Stimulus for sociability and intergenerational dialogue.
3. Local empowerment through involvement of communities.
4. Stimulus for networking between local communities, companies and institutions.
5. Compatibility of tourism activities with the lives of local people.

**CULTURAL SUSTAINABILITY**
[88–90,94]

1. Enhancement of the history and culture of the territory.
2. Development, improvement and enhancement of the territory's image.
3. Stimulus for cultural exchange.
4. Expanding the offer of cultural and recreational activities.
5. Increasing residents' sense of pride and belonging.

**ENVIRONMENTAL SUSTAINABILITY**
[88–90]

1. Contribution to the maintenance of natural resources and the local ecosystem.
2. Stimulating the awareness of local communities and tourists regarding environmental values.
3. Raising awareness and educating local entrepreneurs on ecological transition issues.
4. Awareness of environmental issues related to the upkeep and restoration of historical and cultural heritage.
5. Involvement of local communities in activities related to the environmental recovery and revitalization of the rural heritage and its setting for tourism usage.

## 3. Materials and Methods

Within the theoretical framework provided by the scientific literature on the topic (Section 2), the research aimed to understand the contribution of rural heritage to the four dimensions of sustainability (economic, social, environmental and cultural) for tourism development in rural areas.

In order to answer the research questions identified in Section 1, a qualitative approach was adopted. This methodology allows the investigation of the dynamics of a specific phenomenon in an open and in-depth manner and is often considered best suited to answer exploratory research questions and to capture the complexity of a phenomenon [95–97].

Specifically, the research adopted an exploratory single case study [5,6,98]. This methodology allows a holistic and detailed analysis of the case study embedded in its context and with which it interacts. It helps broaden knowledge of the phenomenon by gathering information about particularly interesting variables from various sources [99].

The chosen case study was considered particularly significant for our research questions as it enhances rural and place-specific cultural heritage, opening up potential opportunities for sustainable tourism development. Therefore, it is an interesting example of how rural heritage contributes to sustainable tourism development.

### 3.1. Motivations and Objectives of the "Marche Food and Wine Memories" Project

"Marche Food and Wine Memories" is a project promoted by the CiùCiù winery, conceived and implemented by the start-up i-strategies, both based in Offida, a small village in the Marche region (Italy). As one of the winery's owners stated, the project stems from the desire to save intangible cultural heritage that is part of the company's family and territory. The company was founded in 1970 by a former sharecropper who decided to buy the land he had, until then, cultivated for a landowner. "The project also arises from the desire to create added value for our brand and products by rediscovering the food and wine roots of our territory" (interview no. 1). Thus, the project aims to safeguard and enhance the rural cultural heritage linked to food and wine in the southern Marche region by collecting the oral memories of the last witnesses of sharecropping, considered a distinctive aspect of local culture and regional identity.

The Italian word *mezzadria* (sharecropping) derives from the Latin "to divide into two halves" and refers to the principle whereby the landowner and the sharecropper family share the produce of the land in two equal portions.

Sharecropping was a rural system that characterized central Italy for centuries. The sharecropping system originated in Tuscany in the 9th century, but began to spread between the 14th and 15th centuries. Although it also reached the United States and France, its greatest penetration was in central Italy, where it survived for over five centuries. Indeed, sharecropping ended between the 1960s and 1970s, due to industrial development and the consequent exodus from the countryside. In the Marche region, it stopped even later, in the 1970s and 1980s [100,101].

The long history of this rural system has left an important legacy in the Marche region in both tangible and intangible cultural heritage. For example, the former is represented by the rural landscape and characteristic farmhouses, while the latter is found in the work ethic, crafts and folklore of the region's inhabitants and, last but not least, in the food and wine.

The "Marche Food and Wine Memories" project has four related objectives (Figure 2):

1.  To create an archive of the oral memories of former sharecroppers as an essential part of the intangible cultural heritage of the Marche region. For this purpose, former sharecroppers were interviewed to collect and preserve their memories.

2.  To save and promote the food and wine heritage of the Marche region: For centuries, wine, olive oil, pasta, fruit and vegetables, along with the limited use of meat, was the daily diet of the sharecroppers of central Italy. These customs were analyzed scientifically in the Seven Countries Study [102], a research project started in 1958 by professor Ancel Key at the University of Minnesota. The study not only identified the so-called Mediterranean diet for the first time, but also documented its protective role against coronary heart disease and other morbid conditions. Part of the research in Italy concerned Montegiorgio, a small village in the Marche region, and its rural population. The project rediscovers and safeguards food heritage and traditions as an authentic, sustainable and inclusive nutritional model. Thus, the objective is to save, promote and transmit to future generations the food and wine memory of former sharecroppers, an essential feature of the culinary heritage of the Marche region and the steward of a safer, healthier and more sustainable food future.

3.  To create added value for the company: local and corporate cultural heritage can be a marketing lever for "Made in" companies, especially those operating in the food and wine sector [103,104]. Indeed, cultural heritage can be an important identity resource for companies seeking a competitive advantage and position in the global market [105–108].

4.  To promote cultural and food-and-wine tourism in the Marche region: the project also includes guided and gamified tours of typical sharecroppers' houses, where the stories collected from the oral memories of former sharecroppers are told to visitors. The experience ends with a tasting of local wines at the CiùCiù company showroom.

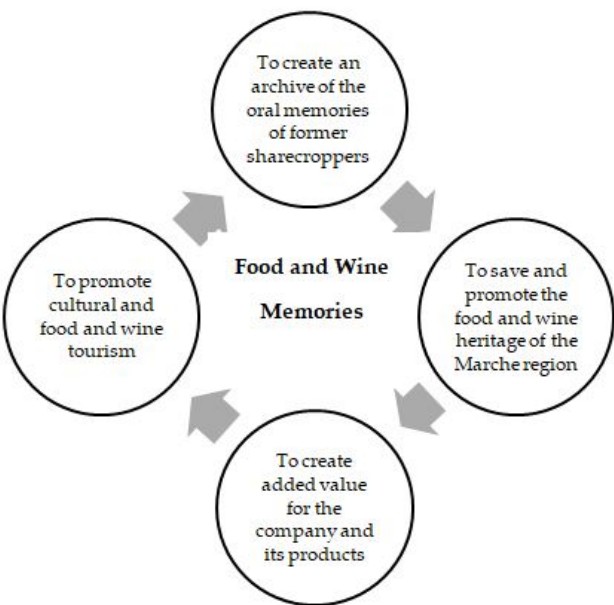

**Figure 2.** Objectives of the project. Source: own elaboration.

*3.2. Project Implementation*

The project started in 2018 and was developed in several phases (Figure 3).

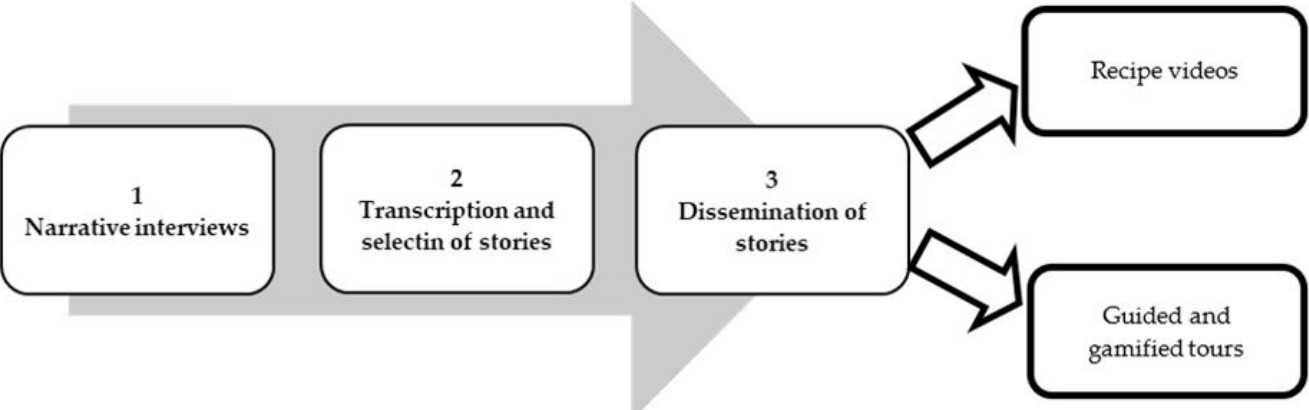

**Figure 3.** Project implementation Source: own elaboration.

The first phase involved interviewing former sharecroppers. Food and wine memories were collected through narrative interviewing [109,110]. The narrative interview is a research methodology that aims to gather an account of lived experiences that is not a collection of answers to questions, but a spontaneous narration, uninterrupted by the researcher's questions. The interviewees, or rather the narrators, thus provide a first-person account of their life or certain aspects of it, and the researcher can consequently work on a complete narrative. This technique is not intended to collect exact data, but rather to encourage witnesses to express, in their own words, even in dialect, their personal experiences.

i-strategies collected about 100 interviews involving people from 17 villages located in the province of Ascoli Piceno (Acquaviva Picena, Appignano del Tronto, Ascoli Piceno, Castel di Lama, Castorano, Colli del Tronto, Comunanza, Cupra Marittima, Grottammare, Monsampolo del Tronto, Monteprandone, Offida, Pedaso, Ripatransone, Rotella, San Benedetto del Tronto and Spinetoli), mainly in the south of the region, amounting to almost 130 h of recordings.

The second phase was transcribing the interviews and selecting the most interesting stories. The selection was based on three criteria:

- Authenticity: stories linked to local history, identity and traditions;
- Driving values: stories as vehicles of values, such as sustainability, food waste reduction and social inclusion;
- Feelings: stories that emphasize the cultural and anthropological aspects of a recipe, the life of sharecroppers and the emotions linked to their oral memories.

The third stage is the dissemination of the collected materials. The selected stories were disseminated in two different ways, online and onsite.

In both cases, the tool used is storytelling, which Serrat [111] (p. 839) defines as "the vivid description of ideas, beliefs, personal experiences and life-lessons through stories or narratives that evoke powerful emotions and insights". Storytelling is a key medium in this project, as it can play an important role in saving and disseminating cultural heritage [112]. Indeed, storytelling can be useful for disseminating cultural heritage in several ways [113–117]:

- Education: storytelling is particularly useful as it helps people understand cultural heritage clearly by giving it purpose and context;
- Engagement: attaching cultural heritage to emotions increases the likelihood of connecting with audiences and creating impact;
- Social cohesion: storytelling brings people together and builds community and social cohesion;
- Cultural heritage protection: storytelling helps create an understanding of the value of tangible and intangible cultural heritage and can lead to greater attachment and protection.

Therefore, part of the project uses digital storytelling and is promoted on the website and social media pages of CiùCiù winery. In fact, one chef tried to modernize traditional recipes taken from the oral memories of sharecroppers and give them a name that encompassed emotions. One example is "tedious tagliolini", inspired by the story of a former sharecropper who, when asked what his snacks were, complained that he ate tagliolini too often. Indeed, sharecropper families had limited resources and flour and eggs were staple ingredients. Hence, egg pasta (tagliolini) was often served, even at breakfast. These interviews enabled the recovery of 40 local recipes.

Videos of the recipes were made by a professional video-maker and shared on CiùCiù winery's social media pages and website. The videos last about one minute. As the images of the chef preparing the dish scroll by, the original voice of the former sharecropper tells her/his own story related to that dish. The recipes are published on the winery's website mostly in step with the seasonality of produce and ingredients.

Another part of the project includes guided tours of sharecropping houses, where this rural system and its tangible and intangible heritage are presented through the life stories collected onsite. Recently, gamified tours have also been tested, especially with students. The gamified tour is structured as a competition in which visitors are divided into teams. After reading the stories based on the memories collected from the former sharecroppers who lived there, visitors are asked to answer questions to collect points. The aim is to experience the intangible heritage of the Marche region through authentic local stories while having fun [118].

In both tours, storytelling is used to convey the history of sharecropping in the Marche region through the voices and faces of those who were part of this rural system, while at the same time actively and emotionally engaging the tour participants.

### 3.3. Material Collected

In order to explore the selected case study, various materials were analyzed (Table 3).

**Table 3.** Material collected (source: own elaboration).

| Material from the "Marche Food and Wine Memories" Project | |
|---|---|
| Publications | Project publications, reports |
| Videos | Video recipes, video reports, promotional videos |
| Online materials | Official website, social networks |
| Pictures | Pictures taken during visits to the sharecroppers' houses |
| Participant observation | Visits to the sharecroppers' houses |
| Internal statistics | Number of visitors taking part in guided and gamified tours |
| Interviews | Local key informants |
| Secondary data | Regional statistics |

We collected materials produced for the project, including project publications, reports and online materials, such as websites and videos. We also analyzed visual data, such as pictures taken during the visits, to document the structure of the tours and communication tools.

More specifically, i-strategies produced a short report on the project in Italian and English. Furthermore, as the start-up is a partner in European projects dealing with heritage marketing and storytelling, documents produced within the framework of these projects are also available. In this case, too, these documents provide information about certain aspects of the project, such as gamified tours.

The online material is very rich. The company has an online presence with a website and social media pages. This allowed an analysis of how and what it communicates on these channels. The winery's social media platforms (Facebook and Instagram) feature videos filmed to safeguard traditional sharecropping recipes. The social media pages link to the winery's website where an entire section is dedicated to "Antiche ricette" ("Ancient recipes", https://www.ciuciutenimenti.it/antiche-ricette/, accessed on 29 October 2022). In addition to videos, the website contains a brief description, the ingredients list, method and recommended wines. In addition, the i-strategies YouTube channel (https://www.youtube.com/channel/UCJK6vFblIXxxELvkcE-Gnmg, accessed on 29 October 2022) provides videos reporting activities carried out within the project. It features a promotional video of the project, the recording of an online roundtable where experts from different institutions (UNESCO, FAO, Maison des Cultures du Monde-Centre français du patrimoine culturel immaterial, Universidade Europeia of Lisbon and NEOMA Business School) were invited to discover and evaluate the project and finally, two videos presenting the work carried out with students from the University of Macerata (Italy) and the University of Heilbronn (Germany) during the experiential activity "Intangible cultural heritage and sustainable tourism development in the Marche region, Italy" organized with the two universities.

Finally, photos and videos taken during the guided and gamified tours are available.

Primary and secondary data were collected in addition to the materials created for the project.

Among the primary data, we had access to information on visitors who took part in guided tours in 2018 and 2019. The activity was suspended in 2020 and 2021 due to the COVID-19 pandemic. Data for 2022 are still partial.

In addition, five semi-structured interviews were conducted between September and October 2022 and involved key informants from the wine company promoting the project, the local administration and local entrepreneurs working in the hospitality and cultural sectors. The design of the interviews was based on the literature review, with a focus on the four pillars of sustainable development and the variables identified as potentially contributing to sustainable tourism development in rural areas. The interviews were recorded, transcribed and analyzed manually.

The purpose of the interviews was to understand:

- Whether knowledge of the project is widespread among various actors operating in the same area;
- The strengths of the project and its contribution to sustainable tourism development and local development;
- The weaknesses of the project, and therefore the areas for improvement.

This investigation is useful for understanding the managerial implications for sustainable tourism development in rural areas and, thus, the contribution of rural heritage.

Finally, on two separate occasions, we conducted participatory observations during gamified tours. The first observation was conducted on 28 April 2022 during a visit organized for the international partners on a European project. The second was conducted on 12 May 2022 during a visit organized for students of the University of Macerata and the University of Heilbronn. During these observations, the structure of the tours and the techniques for presenting the collected heritage were examined.

As secondary data, we collected data from regional statistics on arrivals and presences in the municipality of Offida from 2017 to 2021.

## 4. Results

Examination of the materials collected provides a holistic and in-depth analysis of the case study.

Analysis of the publications suggests that, as a research project, "Marche Food and Wine Memories" aims to create an impact on the territory and particularly on the cultural and social aspects of sustainability. When focusing on the cultural dimensions identified in the theoretical framework, the project collects, saves and enhances local heritage that would otherwise have been lost. This heritage is absolutely place-specific and, therefore, an identity resource for the territory. The project has also restored dignity to the memory of former sharecroppers, increasing the sense of pride and belonging, not only for those who lived and told their stories, but also for their families and those who find elements of their own family histories in those narratives. In terms of social sustainability, the project has certainly stimulated intergenerational dialog. Indeed, by reviving interest in the sharecropping heritage, it has also shed light on some of the traditional values of this rural society that should be reconsidered today, such as sociability, not wasting food and frugality.

Analysis of online materials, videos and photos shows that communication on the web is very effective and can enhance the project and the activities already carried out. It will be interesting to study how many people the posts on social media and the video recipes have reached and what kind of reaction they have provoked.

Participatory observations during the visits allowed us to improve our understanding of the structure of the tours and the techniques used to present the heritage gathered from the memories of former sharecroppers. Although the gamified tour is still at an experimental stage, a number of observations can be made. The tours are well structured and well designed. Reading a story with illustrations and questions is an effective way to engage participants. One way to improve the experience could be to use digital tools: currently, all communication tools are paper-based. In addition, the environment should be equipped to meet participants' needs, e.g., by providing shaded areas during tours. Rural heritage is presented effectively during gamified tours. The thoughtfully selected stories successfully evoke a time that now seems to be very distant. Moreover, one of their strengths is that they are told in the place where they occurred. However, the focus is mostly on the stories and the intangible heritage they reveal, and the surrounding tangible heritage is sometimes neglected. It would be good to devote space to rural evidence from the surrounding area, such as rural houses and the landscape that has been considerably shaped by sharecropping culture. Other comments concern access to the visits. So far, visits have been offered alongside other events. No information is provided about booking a visit, making it an "invitation only" event. Important targets are high schools (especially

those specializing in catering and hospitality) and universities (tourism, cultural heritage and wine marketing). Still, it would be interesting to open up this activity to a wider public.

Data on visitors, especially from the pre-pandemic years, can give us some additional insight. As shown in Figure 4, the number of participants on guided tours increased before the COVID-19 pandemic. In 2018, 212 people (103 foreigners and 109 Italians) took part in the guided tours, and in 2019, the number was 305 (163 foreigners and 152 Italians). After a two-year stoppage (2020 and 2021), visits resumed slowly. For 2022, the graph shows data up to the summer (55 participants, including 18 Italians and 37 foreigners).

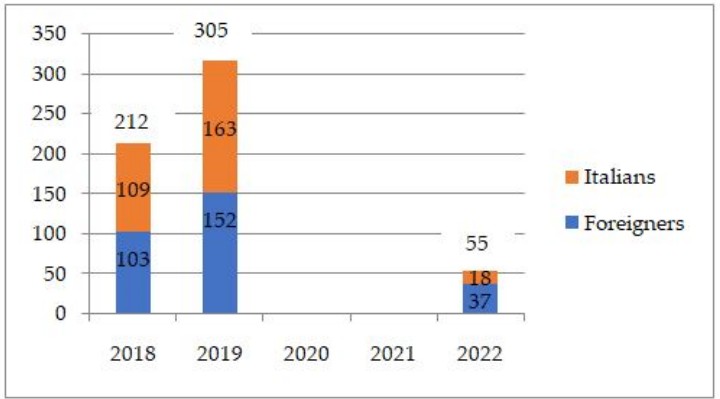

**Figure 4.** Number of visitors to the sharecroppers' houses. Source: own elaboration.

Two observations can be made from the data on visitors' country of origin (Figures 5 and 6). First, this project holds great interest for foreign visitors, even though at least half of participants in 2018 and 2019 were Italian. British visitors are the most represented among them. This is an interesting aspect, as many British people have moved to the Marche region. This project can give them a better understanding of the culture of the place they live in and help them integrate culturally.

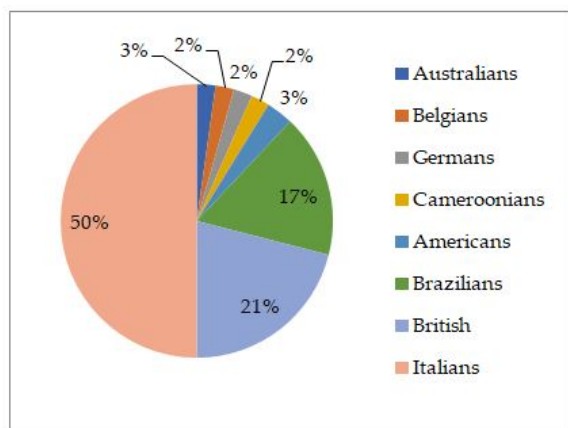

**Figure 5.** Visitors' countries of origin (2018). Source: own elaboration.

Further considerations should be made when these data are compared to regional statistics (https://statistica.regione.marche.it/Marche-in-Numeri/Turismo, accessed on 29 October 2022) on arrivals in Offida. As shown in Figure 7, arrivals in Offida were on the decline before the pandemic. There was a slight recovery in 2021, and data for 2022 are not yet available. Even though tourism was declining in 2017–2019, it is undeniable that the pandemic could be an opportunity for future tourism development. COVID-19 has accelerated some trends that were already underway, such as the decline in mass tourism in big cities and the growing preference for less well-known, less frequented and more

authentic places that allow the tourist to temporarily feel like a citizen. This makes rural heritage a key resource for tourism development in the coming years.

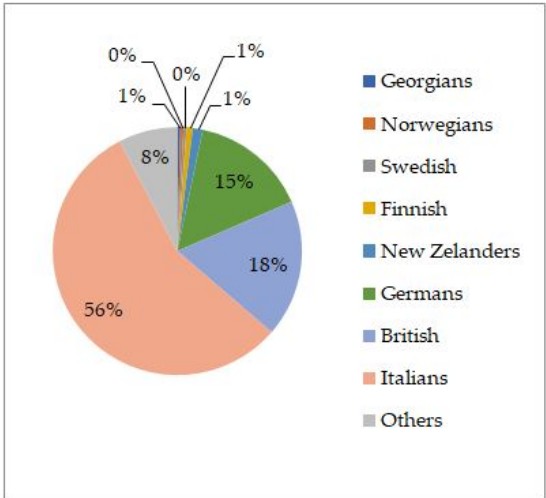

**Figure 6.** Visitors' countries of origin (2019). Source: own elaboration.

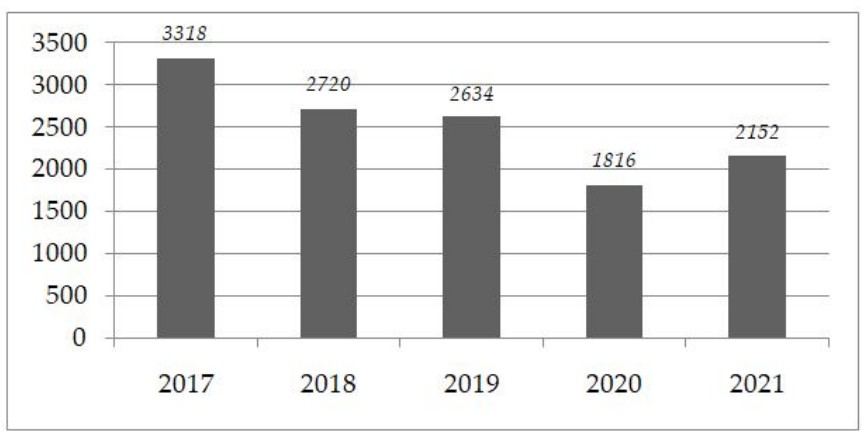

**Figure 7.** Total arrivals in Offida (Italy) from 2017 to 2021. Source: own elaboration.

## 5. Discussion

"Marche Food and Wine Memories" is a project that aims to safeguard and enhance the rural cultural heritage linked to food and wine through the oral memories of the last witnesses of sharecropping as a distinctive element of local culture and regional identity. The analysis of the various collected materials allowed us to understand the main strengths and weaknesses of the project and to reflect on the project's current contribution to the different dimensions of sustainability.

The above observations regarding the role of cultural heritage in sustainable development (Table 2) are the basis for a number of considerations.

From an economic point of view, local companies in the field of tourism and hospitality stated that the visits organized within the project increased their customer numbers. Indeed, one of the interviewees reported that, of the people who came to Offida for this project, "some came back, others sent someone else" (interview no. 3). The project is welcomed because it attracts tourists, enhances the place's image, and boosts the local economy. However, so far, visits have only been offered occasionally, thus preventing the attainment of the economic benefits that the project has the potential to create. Under the current setup, the project cannot boost local entrepreneurship, create new job opportunities or attract external investments. Another weakness that emerged from the interviews is that the project alone cannot create sustainable tourism in the area. A network is needed of the

various local actors working towards a common goal. In this regard, one of the interviewees stated: "the project has potential; it is certainly a resource that can promote sustainable tourism, but on its own, it is unlikely to be successful. Collaboration between different local public and private actors is needed for developing a more structured and effective idea" (interview no. 5). Therefore, from a managerial point of view, greater involvement and collaboration with the social, economic and local administrative actors could be the key for rural heritage to truly contribute to sustainable tourism development.

From a social point of view, the project's potential for an intergenerational and cultural exchange with visitors is recognized. Indeed, storytelling can stimulate curiosity about one's family history, perhaps linked to sharecropping. This opens a dialogue between different generations that can strengthen relationships and sociability and stimulate new ideas. Moreover, especially for foreign tourists, "it is like coming into contact with a distant and exotic civilization that they do not know and which they are immediately curious about" (interview no. 1). This leads to exchange and openness towards others. However, what is well expressed in the documents and materials produced for the project is not well received by local stakeholders and the local community. Generally, the project is not well known at local level, and when it is, only superficially. From the point of view of marketing, there should be better communication of the project with the local community because "informing locals of this project can have a resonating effect in the area; otherwise, the risk is that the project remains somewhat isolated and an end in itself" (interview no. 2). Indeed, a better understanding of the goals and the work behind them could increase engagement and appreciation among residents and facilitate collaboration. This is fundamental, in terms of sustainable tourism, as the active involvement of the local community would avoid the risk of incompatibility between tourist activities and the lives of locals. Ultimately, how the project is structured and communicated has, to date, failed to produce effects on quality of life or to encourage the creation of networks between local communities, companies and institutions. At the managerial level, improving these aspects will be crucial for sustainable tourism.

Culturally speaking, the rescue and enhancement of a heritage that had previously gone unnoticed and would otherwise have been lost are certainly recognized as a strength. This has had the consequence of restoring the dignity to the rural heritage that living witnesses had not considered worthy of attention and rediscovering the family roots of former sharecroppers. The project has often been defined as "far-sighted". Moreover, the re-evaluation of the sharecropping heritage is recognized as an additional asset for the local cultural offer, which already boasts a remarkable heritage of history and art, craft traditions and food and wine. In fact, "the project adds a further piece to the culture of Offida, and this could be an attraction for different types of tourists" (interview no. 4). The project enables further differentiation of the offer and potentially attracts new visitors. From a managerial point of view, digital technologies can support further development of the project. One respondent suggested that "to ensure the project has an economic impact, it is necessary to find a modern way whereby those who come can enjoy this heritage, even independently. Digitalization could make these memories more material" (interview no. 5).

From an environmental point of view, the project is recognized as having the potential to increase awareness of environmental values among local communities and tourists alike. "The project helps us to reflect on the way the territory, society, the economy has developed over time and perhaps leads us to reassess certain values and practices deemed outdated but, in reality, very current" (interview no. 4). In particular, the interviewees emphasized its value for the younger generations, who consider rural culture to be very remote. "This project can promote environmental sustainability, but needs to be embraced by more people to be effective" (interview no. 2). There is again a need to work in concert with others and to involve and train stakeholders and the local community to be able to provide truly sustainable services that are also environmentally sound.

The project's current contribution to sustainability is summarized in Table 4.

**Table 4.** Contribution of the "Marche Food and Wine Memories" project to sustainability. Source: own elaboration.

| ECONOMIC SUSTAINABILITY | |
|---|---|
| *Theory* | *Project (today)* |
| 1. Source of income for the tourism and hospitality sectors | An occasional source of income |
| 2. Source of income for the tourism and hospitality-related sectors | An occasional source of income |
| 3. Encouragement for local entrepreneurship. | Not currently |
| 4. Source of attraction for external investment | Not currently |
| 5. Employment opportunities for residents | Not currently |
| SOCIAL SUSTAINABILITY | |
| *Theory* | *Project (today)* |
| 1. Enhancement of quality of life | Not currently |
| 2. Stimulus for sociability and intergenerational dialogue | Through interviews and emphasis on sharecropping values |
| 3. Local empowerment through involvement of communities | Not currently |
| 4. Stimulus for networking between local communities, companies and institutions | Not currently |
| 5. Compatibility of tourism activities with the lives of local people | Not detectable with our data |
| CULTURAL SUSTAINABILITY | |
| *Theory* | *Project (today)* |
| 1. Enhancement of the history and culture of the territory | Collecting, saving and promoting a local heritage that would otherwise have been lost |
| 2. Development, improvement and enhancement of the territory's image | Not currently |
| 3. Stimulus for cultural exchange | Between different generations and cultures |
| 4. Expanding the offer of cultural and recreational activities | Not currently |
| 5. Increasing residents' sense of pride and belonging | Restoring dignity to rural heritage |
| ENVIRONMENTAL SUSTAINABILITY | |
| *Theory* | *Project (today)* |
| 1. Contribution to the maintenance of natural resources and the local ecosystem | Not currently |
| 2. Stimulating the awareness of local communities and tourists regarding environmental values | Partially: it is embedded in the rural heritage, but it needs to be enhanced |
| 3. Raising awareness and educating local entrepreneurs on ecological transition issues | Not currently |
| 4. Awareness of environmental issues related to the upkeep and restoration of historical and cultural heritage | Not currently |
| 5. Involvement of local communities in activities related to the environmental recovery and revitalization of the rural heritage and its setting for tourism usage | Not currently |

## 6. Conclusions

With a view to making a theoretical contribution to the debate on tourism sustainability, this paper has investigated the possible contribution of cultural heritage to sustainable tourism development in rural areas. The innovation of the research consists in having suggested a connection between the three dimensions of cultural heritage value and the four

dimensions of sustainability (Figure 1). After recognizing the close relationship between these aspects, a framework has been provided (Table 2) to guide the analysis of a case study.

The case study was chosen because it is considered explanatory. Furthermore, as already mentioned, studying this phenomenon in the rural context is urgent, in light of the need to reduce territorial imbalances in different areas, as recently highlighted by European regional policies and structural funds. For example, Italy allocated part of the funds that the European Union provided after the COVID-19 pandemic to rural areas, expressly linking the development of these areas to sustainable tourism.

When it comes to the analysis of research results, we can argue that the "Marche Food and Wine Memories" project is currently contributing to cultural sustainability, but its contribution to the other pillars of sustainability should be improved. In fact, cultural sustainability is only the first step in triggering a virtuous circle that benefits social, economic and environmental sustainability. In particular, some fundamental elements emerge from the areas of improvement: involvement of the local community, involvement and training of local stakeholders and networking between the different actors in the territory. Moreover, from an organizational point of view, the activities should be planned and managed regularly with the support of professional marketing and management skills. The use of technologies and communication techniques can be a very useful support tool for making the intangible heritage that is enhanced by this type of project more tangible.

To conclude, it is possible to promote sustainable tourism through the enhancement of place-specific cultural (intangible) heritage only if the local community is involved, dialogue and collaboration are created between the various local actors and the right professional skills are involved. Therefore, cultural and human resources are important resources for the development of the territory.

The research examined only one case study. Further investigation should collect quantitative data on heritage-based tourism initiatives to frame their contribution to sustainable development. Moreover, future research should be directed at analyzing the perceptions of tourists.

**Author Contributions:** Conceptualization, M.C., A.R. and G.V.; methodology, M.C. and A.R.; software, A.R.; validation, M.C., C.F. and A.R.; formal analysis, M.C. and C.F.; investigation, C.F. and A.R.; resources, A.R. and G.V.; data curation, A.R. and G.V.; writing—original draft preparation, M.C., C.F., A.R. and G.V.; writing—review and editing, M.C., C.F. and A.R.; visualization, M.C. and A.R.; supervision, M.C.; project administration, M.C.; funding acquisition, M.C. All authors have read and agreed to the published version of the manuscript.

**Funding:** This research was co-funded by the University of Macerata and the Marche Region (POR Marche FSE 2014/2020, Progetto "Dottorato Innovativo"—Borse di studio per dottorato di ricerca per l'innovazione del sistema regionale, 2020).

**Institutional Review Board Statement:** Not applicable.

**Informed Consent Statement:** Not applicable.

**Data Availability Statement:** The data used to support the findings of this study are available from the corresponding author upon request.

**Acknowledgments:** We wish to thank the anonymous reviewers for their insightful comments and suggestions, which helped us improve the quality of our paper. We also owe special thanks to the editors of the Special Issue on "Sustainable Wine and Beverage Tourism" for accepting our research proposal and supporting its publication.

**Conflicts of Interest:** The authors declare no conflict of interest.

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
