# Peer review of "Enhancing Intangible Cultural Heritage for Sustainable Tourism Development in Rural Areas: The Case of the “Marche Food and Wine Memories” Project (Italy)"

_sustainability, doi:10.3390/su142416893_

Round 1
Reviewer 1 Report
Thanks for giving the opportunity to review the paper titled as Enhancing intangible cultural heritage for sustainable tourism development in rural areas. The case of the “Marche food and wine memories” project (Italy). A This paper is timely, well developed, conducted and well written. It addresses a significant topic likely to be of interest to destination sustainable development in rural areas. Thus, it addresses a significant topic likely to be of interest to Sustainability. Despite all of this, there are several possible revisions as the following:
First, it’s maybe right that author(s) point(s) out enhance intangible cultural heritage for sustainaible tourism development in rural areas. However, However, destination crisis event has negative impacts on tourist’s emotion, attitudes and behavior in turn influence sustainable tourism development (Su, Jia, & Huang, 2022; Su, Pan, & Huang, 2023), especially in the COVID-19 pandemic context. Please author(s) refer(s) these references, and explain why they do not consider these factors. At least, author(s) need mention them, and list these as a limitation.
“How do destination negative events trigger tourists’ perceived betrayal and boycott? The moderating role of relationship quality”. Tourism Management, 92, 104536.
“How does destination crisis event type impact tourist emotion and forgiveness? The moderating role of destination crisis history”. Tourism Management, 94, 104636.
Second, it needs point out the research gaps more clearly in the introduction section.
Third, the influencing mechanism of cultural heritage on sustainable development needs more clearly. Specially, what’s the uniqueness and role cultural heritage in rural areas compared to other areas.
Fourth, overall, the contribution of the paper and the discussion of the contribution should be made more clearly.
Fifth, it need point out the limitations and future research directions more specifically in the end of the paper.
Though the above-mentioned possible shortages, this paper is a high-quality original article. Thus, I recommend author(s) revise(s) the paper according to reviewer’s comments/suggestions.
Author Response
We list here the main changes we made. All the changes are highlighted in yellow in the paper.
The paper has also been submitted to a proofreader for checking the English language.
- First, it’s maybe right that author(s) point(s) out enhance intangible cultural heritage for sustainable tourism development in rural areas. However, destination crisis event has negative impacts on tourist’s emotion, attitudes and behavior in turn influence sustainable tourism development (Su, Jia, & Huang, 2022; Su, Pan, & Huang, 2023), especially in the COVID-19 pandemic context. Please author(s) refer(s) these references, and explain why they do not consider these factors. At least, author(s) need mention them, and list these as a limitation.
- Su, Jia, & Huang, 2022, “How do destination negative events trigger tourists’ perceived betrayal and boycott? The moderating role of relationship quality”. Tourism Management, 92, 104536.
- Su, Pan, & Huang, 2023, “How does destination crisis event type impact tourist emotion and forgiveness? The moderating role of destination crisis history”. Tourism Management, 94, 104636.
Thank you for your comments and the bibliographical references you suggested. We have further explored and developed the relationship between intangible cultural heritage, sustainable tourism development and rural areas within Section 2.3. The role of cultural heritage for sustainable development. We also included a short focus on the role of authenticity in sustainable rural tourism in the previous section (2.2. Sustainable tourism development in rural areas). As for the impact of Covid-19 on tourism, as suggested, we first focused on the negative impact of the pandemic as a “destination crisis event” on tourist attitudes and behaviors and then we quickly explored how Covid-19 impacted tourism and opportunities that could result for rural destinations also from a sustainability perspective.
- Second, it needs point out the research gaps more clearly in the introduction section.
Thank you for highlighting this limitation. In the introduction, we specified how our research tries to fill current gaps. In the conclusion section, we also highlighted the theoretical contribution brought by the paper.
- Third, the influencing mechanism of cultural heritage on sustainable development needs more clearly. Specially, what’s the uniqueness and role cultural heritage in rural areas compared to other areas.
Thank you for your comments. We better developed this topic within Section 2.3. The role of cultural heritage for sustainable development, by focusing on the role of intangible cultural heritage in terms of cultural authenticity.
- Fourth, overall, the contribution of the paper and the discussion of the contribution should be made more clearly.
Thank you for your comments. We have further discussed the results to make them clearer. In particular, we reflected more on the project’s strengths and areas of improvement.
- Fifth, it need point out the limitations and future research directions more specifically in the end of the paper.
Thanks for the helpful suggestion. In the last section, we have discussed the research limitations and possible further research to improve the analysis of this phenomenon.
Though the above-mentioned possible shortages, this paper is a high-quality original article. Thus, I recommend author(s) revise(s) the paper according to reviewer’s comments/suggestions.
Reviewer 2 Report
This is an exciting paper. I believe that the authors are discussing an important topic in Cultural Heritage and Sustainable Tourism. However, the authors need to describe knowledge gaps better.
Table 2 title mentions "the three dimensions" but the table has four dimensions, as from opinion.
The method section provides a good description of how data was collected but not how it was analysed (tools, methods, techniques, procedures, etc.)
In the results section, only some descriptive information is presented. I suggest the authors present also the results of qualitative data.
Finally, I think that the conclusions section felt short and need improvement, namely by stressing the relevance and contribution of the manuscript, as in its current form it is a repetition of previous sections.
Author Response
We list here the main changes we made. All the changes are highlighted in yellow in the paper.
The paper has also been submitted to a proofreader for checking the English language.
- Table 2 title mentions “the three dimensions” but the table has four dimensions, as from opinion.
Thank you for having highlighted this typo. We corrected the title of the table.
- The method section provides a good description of how data was collected but not how it was analysed (tools, methods, techniques, procedures, etc.)
Thank you for your observation. We added some information to clarify how we analyzed the materials, especially interviews.
- In the results section, only some descriptive information is presented. I suggest the authors present also the results of qualitative data.
Thank you for the suggestion. In the discussion section, we added more insights into the content of the qualitative analysis. We also quoted and discussed some qualitative data that emerged during the interviews.
- Finally, I think that the conclusions section felt short and need improvement, namely by stressing the relevance and contribution of the manuscript, as in its current form it is a repetition of previous sections.
Thank you for your comment. We have deepened our reflections in the conclusion section, focusing on the innovation and relevance of our research in the current context, the possible outcomes of this kind of initiative, the limitations and possible further developments.
Round 2
Reviewer 1 Report
The paper has improved geatly. Thus, the reviewer recommend to accept it in present form.
Reviewer 2 Report
Well done.